# Antioxidant-Mediated Modification of Citral and Its Control Effect on Mildewy Bamboo

**DOI:** 10.3390/polym14214652

**Published:** 2022-11-01

**Authors:** Chunlin Liu, Qi Li, Yingying Shan, Chungui Du, Shiqin Chen, Wenxiu Yin, Fei Yang, Yuran Shao, Yuting Wang

**Affiliations:** College of Chemistry and Materials Engineering, Zhejiang A & F University, Hangzhou 311300, China

**Keywords:** citral, degradation properties, antioxidant, modification, bamboo, antimildew

## Abstract

To reduce the oxidative degradation of citral and improve its antimildew performance, citral was modified with natural antioxidants such as tea polyphenols, ascorbic acid, and theaflavin in the present study. Additionally, the effects of these natural antioxidants on the citral degradation rate and DPPH radical-scavenging rate, as well as the effectiveness of antioxidant-modified citral in the antimildew treatment of bamboo were investigated. Ascorbic acid, theaflavin, and tea polyphenols improved the antioxidant performance of citral to some extent, and the tea polyphenols exhibited the best antioxidant performance. When the amount of tea polyphenols added to citral reached 1.0%, the oxidative degradation of citral was effectively prevented. Compared with citral, tea-polyphenol-modified citral could reduce the efficacy of the bamboo antimildew treatment against all four mildews and the effectiveness of the antimildew treatment reached 100%. Citral modification with antioxidants reduced the amount of citral required in the treatment, thereby reducing the treatment cost for bamboo mildew.

## 1. Introduction

As the fastest growing plant globally, bamboo is beneficial because of its biodegradability, renewability, high tensile strength, toughness, and low cost [1]. Bamboo and its products are widely used in construction, furniture manufacturing, and gardening [2,3,4,5]. However, bamboo is highly prone to mildew and surface contamination. The damage caused is so profound that bamboo loses its use value and is discarded, resulting in a significant waste of resources and economic loss [6,7]. Therefore, research on antimildew agents to protect bamboo is vital and urgently required. Bamboo wood mildew prevention has gradually become a research hot spot.

The research and application of natural antimicrobial agents have increased considerably. Among these agents, citral, a green, non-toxic natural antimicrobial agent mainly derived from *Litsea cubeba* essential oil, has a broad-spectrum antimicrobial activity and a strong lemon fragrance [8,9,10]. It has great application prospects in bamboo antimildew treatment. Zhang et al., [11,12] showed that when the citral concentration used for bamboo treatment reached 200 mg/mL, the efficacy of control on common mildews of bamboo was 100%. However, citral contains carbon–carbon double bonds and carbonyl groups, which are chemically unstable and susceptible to oxidative degradation by external factors such as oxygen, temperature, and pH, resulting in reduced antibacterial properties and thus limiting the application of citral [13,14,15,16,17]. To improve citral stability, Ju [8], Chang [18], Qiu [19], Wang [20], an Lu [21], et al., as domestic and Arnon-Rips [22], Mendes [23], and Sharma [24] et al., as foreign scholars have used microcapsules to encapsulate citral and to encapsulate citral with antioxidants. Among these antioxidants, natural plant antioxidants have become the focus of antioxidant research because of their strong antioxidant properties, wide range of sources, high affinity, and high safety [25,26,27,28]. Only a few scholars have studied citral modification by natural antioxidants. Liang et al. [26] used plant extracts to inhibit the antioxidant activity of citral off-flavor formation. The results showed that plant phenolic extracts significantly inhibited the oxidative degradation of citral and thus the production of odorant substances. Shi et al. [29] investigated the antioxidant, antibacterial, and antitumor activities of ε-poly-lysine (ε-PL) and citral alone and in combination. The combination of ε-PL and citral significantly inhibited the growth of bacterial strains, enhanced the stability of the antimicrobial qualities of citral, and reduced citral oxidative degradation, thus enhancing the antimicrobial activity of citral. However, no study has investigated the antibacterial properties of antioxidant-modified citral against bamboo mildews. Therefore, this study discusses the efficacy of natural antioxidants on the antimildew properties of citral. The research results lay the foundation and provide a theoretical reference for the promotion and application of natural antioxidant modified citral as an antimildew agent for bamboo.

## 2. Materials and Methods

### 2.1. Materials

Citral (C, 97%) and tea polyphenols (TP, 97%) of analytical purity were purchased from Shanghai Macklin Chemical Reagent Co., Ltd. (Macklin, Shanghai, China) Theaflavins (TF, 97%) of analytical purity were purchased from Jiangsu Hongda Biotechnology Co., Ltd. (Hongda, China) Bamboo strips made of Moso bamboo timber were processed into bamboo slices of size 50 mm × 20 mm × 5 mm (length × width × thickness) without bamboo nodes and with a moisture content of approximately 10%. The bamboo strips were purchased from Xizhuyuan Bamboo Products Factory in Zhenghe County, Fujian Province, China. The mildews used in the test were *Penicillium citrinum* (*PC*), *Trichoderma viride* (*TV*), *Aspergillus niger* (*AN*), and mixed mildew (HUN, containing a mixture of *PC*, *TV*, and *AN* in equal proportions).

### 2.2. Antioxidant-Modified Citral Standard Curve

A propylene glycol solution (pH: 6.0) was prepared at room temperature [29]. The appropriate amount of citric acid was mixed with the propylene glycol solution and stirred in a heat-collecting magnetic stirrer until the citric acid dissolved. The pH of the solution was measured using a micro pH meter until the pH was 6.0. Then, 1.0 g of citral and 0.01 g of natural antioxidants (theaflavin, tea polyphenols, and ascorbic acid) were weighed [30], and 10 mL (10% of the total volume of the solution) of deionized water and the appropriate amount of propylene glycol with a pH of 6.0 were added (see in Appendix A). The solution to be measured was stirred with a magnetic stirrer at 100 rpm for a 0.5 h and then poured into a 100 mL volumetric flask and fixed with anhydrous ethanol to obtain three concentrations of tea-polyphenol–citral (TP-C), ascorbic-acid–citral (ASC-C), and theaflavin–citral (TF-C) at 10 mg/mL; finally, the aforementioned citral solutions were diluted with anhydrous ethanol to obtain natural antioxidant-modified citral solutions of 1, 2, 3, 4, and 5 µg/mL concentrations. Pure citral was used as the control. Absorbances of the aforementioned dilutions were measured using a UV-1800 UV–Vis spectrophotometer in the 400–200 nm wavelength range at the maximum absorption wavelength λ max. Each parallel sample was measured three times, the results were averaged, and the standard curves of different natural antioxidant-modified citral solutions were plotted on Origin.

### 2.3. Effect of Antioxidants on the Degradation Performance of Citral

The effects of antioxidant type, additional amount, and solution concentration on the degradation performance of antioxidant-modified citral were investigated. Propylene glycol solution of pH 6.0 was prepared according to the method described in Section 2.2. Then, 2 mL (10% of the total volume of the solution) of anhydrous ethanol and 2 mL (10% of the total volume of the solution) of deionized water were added to the test tube with a pipette, followed by the addition of 3.5 g of citral and 0.035 g of natural antioxidant. The solution to be measured was made up to 20 mL with the aforementioned propylene glycol solution. The solutions to be tested were stirred with a magnetic stirrer at 100 rpm for 0.5 h, and, finally, the solutions to be tested were prepared with TP-C, ASC-C, and TF-C at 175 mg/mL and natural antioxidants at 1.0%. The pH values of the solutions were measured using a micro pH meter with a pH error of ± 0.1. Similarly, the antioxidant-modified citral was prepared as described above at a 150 mg/mL concentration with the addition of 0.8% natural antioxidants and at a 125 mg/mL concentration with the addition of 0.6% natural antioxidants (see in Appendix A). Natural antioxidants were added at a rate of 1% of the mass of citral. The antioxidant-modified citral solutions were stored in an artificial climate chamber at 28 °C and 60% ± 5% humidity, and the citral degradation rates were determined after 1, 7, 14, and 28 days of storage to study the effect of antioxidants on the degradation performance of citral. The citral degradation rate was calculated according to Equation (1).
DR (%) = (m0 − mi)/m0 × 100%(1)

In the equation, DR is the degradation rate of antioxidant-modified citral, %; m0 is the mass of antioxidant–citral solution before storage; and mi is the mass of antioxidant-modified citral solution 28 days after storage.

### 2.4. Determination of Antioxidant-Modified Citral DPPH Radical-Scavenging Rate

1,1-Diphenyl-2-trinitrophenylhydrazyl radical 2,2-diphenyl-1-(2,4,6-trinitrophenyl) hydrazyl (DPPH) solution was prepared [31]. Then, 0.01 g DPPH was dissolved with the appropriate amount of anhydrous ethanol. The solution was poured into a 100 mL volumetric flask and shaken well. Then, the solution was poured into a 100 mL volumetric flask and its volume was fixed by adding anhydrous ethanol solution and shaking the mixture. DPPH solution (20 mL) was added to a 100 mL volumetric flask, diluted 5 times with anhydrous ethanol solution, and placed at 4 °C (see in Appendix A). First, 10 µL of each of TP-C, ASC-C, and TF-C solutions stored for 28 days in the modified citral concentration assay was taken in a test tube by using a pipette, and 5 mL of anhydrous ethanol solution was added to the tubes and mixed well. Then, 100 µL of each of the aforementioned mixed solutions was taken in a test tube, and 10 mL of anhydrous ethanol solution was added and mixed well. Similarly, the pure citral was diluted as mentioned above. The solutions to be tested were prepared in the following amounts: A0 = 3 mL of DPPH solution + 1 mL of pure citral dilution to be tested and Ai = 3 mL of DPPH solution + 1 mL of antioxidant-modified citral dilution to be tested. The solutions to be tested were allowed to stand for 30 min. Finally, the absorbance of the solutions was measured at a wavelength of 517 nm by using a UV-1800 UV–Vis spectrophotometer. The absorbance and the free-radical-scavenging rate of DPPH were calculated according to Equation (2).
DPPH (%) = (A0 − Ai)/A0 × 100%(2)

In the equation, DPPH is the scavenging rate of free radicals in citral solution, %; A0 is the absorbance of pure citral DPPH solution; and Ai is the absorbance of antioxidant-modified citral DPPH solution.

### 2.5. Fourier-Transform Infrared Spectroscopy of Antioxidant-Modified Citral

First, the antioxidant-modified citral solution was prepared using the antioxidant-modified citral preparation method reported in an aforementioned study, and the solution was stored at 28 °C for 28 days. Second, potassium bromide powder was ground into a fine powder by using a mortar and dried thoroughly using a heat lamp. Then, the powder was pressed into thin slices for preparation, and the antioxidant-modified citral sample solution was added dropwise onto the pressed slices by using a spotting capillary column to fill the sheet with the sample solution. Finally, this loaded sheet was placed into an IR Prestige-21 Fourier-transform infrared spectrometer (FT-IR) for the characterization and analysis of the molecular structure of the antioxidant-modified citral sample.

### 2.6. Efficacy of Antioxidant-Modified Citral against Bamboo Mildew

The antioxidant-modified citral solution prepared using the better antioxidant-modified citral preparation process was used to investigate the effect of concentration on the effectiveness of antimildew treatment of bamboo chips. Antioxidant-modified citral solutions with concentrations of 150, 175, and 200 mg/mL and pH 6.0 were prepared according to the method described in 2.3, with a pH error not exceeding ± 0.1. Blank bamboo slices were prepared without adding any agent. According to the process parameters of Zhang et al. [11] for pressure impregnation of the bamboo material, bamboo slices were pressure impregnated with the aforementioned concentrations of the antioxidant-modified citral solution and were referred to the relevant provisions of the national standard “Test method for the effectiveness of antimildew agent against wood mildew and discoloration fungus” (GB/T 18261-2013) [32]. *PC*, *TV*, *AN*, and HUN were evenly coated on a PDA plate medium, and the pressure-impregnated bamboo slices were placed on U-shaped glass rods, sealed with a sterile sealing film, and placed in an artificial climate chamber at 28 ± 2 °C and a relative humidity of 85 ± 5% for the mildew test. The bamboo slices were observed and recorded daily for infection by *PC*, *TV*, *AN*, and HUN to determine the infection value of the bamboo slices (Table 1). The effectiveness of the treated bamboo slices against bamboo mildews was calculated according to Equation (3).
(3)$$E\;(\%)=(1\;-\;\frac{D_{1}}{D_{0}})\;\times \;100\%$$

In the equation, E is the efficacy of the control, %; D_1_ is the mean infection value of antioxidant-modified citral solution-treated bamboo slices; and D_0_ is the mean infection value of control bamboo slices.

## 3. Analysis of Results

### 3.1. Preparation of Standard Curve for Antioxidant-Modified Citral

The UV absorption spectra of antioxidant-modified citral were plotted using Origin software, as shown in Figure 1. The spectra include the UV profiles of pure citral (C), TP-C, ASC-C, and TF-C solutions in the 400–200 nm wavelength range (Figure 1a), and the standard curves of 1–5 µg/mL TP-C, ASC-C, and TF-C solutions at a 238 nm wavelength (Figure 1b–d).

The maximum absorption peaks of TP-C, ASC-C, and TF-C at 238 nm in Figure 1a coincided with the absorption peak of pure citral, which indicated that the TP-C, ASC-C, and TF-C solutions were not affected by natural antioxidants or the citral structure was not damaged. The absorption at this wavelength was consistent with the findings of Peng [33] and Ay [34]. Figure 1b shows the standard curve regression equation of the TP-C solution, A = 0.0803C + 0.0295, R^2^ = 0.99285. Figure 1c shows the standard curve regression equation of the ASC-C solution, A = 0.0858C + 0.0148, R^2^ = 0.99707. Figure 1d shows the standard curve regression equation of the TF-C solution, A = 0.0798C + 0.0319, R^2^ = 0.99472. The fitted correlation coefficient of the aforementioned regression equation, R^2^, was very close to 1, indicating that the concentration of the antioxidant-modified citral had a good linear relationship with absorbance. The experimental error using this standard curve was small, thus meet the experimental requirements. Therefore, this standard curve equation could be used to calculate the citral concentration in the antioxidant performance experiments of antioxidant-modified citral solutions.

### 3.2. Effect of Antioxidants on the Degradation Performance of Citral

#### 3.2.1. Effect of Antioxidant Type on the Degradation Performance of Citral

The antioxidant-modified citral solution was prepared by adding 1% by mass of tea polyphenol, theaflavin, ascorbic acid, and other natural antioxidants to the citral and stirring the solution well with a magnetic stirrer. This solution was stored at room temperature for 28 days, and the citral degradation rate was tested. The results are shown in Figure 2.

As shown in Figure 2, the citral degradation rate in the ASC-C solution was 22.27%, that in the TF-C solution was 15.23%, and that in the TP-C solution was 6.48%. After the modification of citral with ascorbic acid, the citral continued to undergo degradation and could not effectively prevent the oxidative degradation of the citral. On the other hand, the degradation rate of the citral modified with tea polyphenols was lower, which could effectively prevent the oxidative degradation of the citral and thus improve the chemical stability of the citral. This was mainly because ascorbic acid is affected by light and temperature during storage, which easily leads to the production of deoxyascorbic acid, followed by oxalic acid, thereby reducing its antioxidant capacity. Tea polyphenols are less affected by light and temperature during storage and maintain their own antioxidant stability; therefore, tea polyphenols could effectively improve the chemical stability of citral. Accordingly, tea polyphenols are promising natural antioxidants for preventing the oxidative degradation of citral.

#### 3.2.2. Effect of Antioxidant Addition on the Degradation Performance of Citral

The aforementioned results showed that tea polyphenols could effectively prevent the oxidative degradation of citral. Based on this finding, the effect of tea polyphenol addition on the degradation performance of citral was investigated, and the results are shown in Figure 3.

As shown in Figure 3, the citral degradation rate decreased significantly with an increase in the tea polyphenol content. With the addition of tea polyphenol at 0.6%, the citral degradation rate was 13.95%; with 0.8% tea polyphenol, the degradation rate was 10.22%; and with 1.0% tea polyphenol, the degradation rate was 6.48%. Compared with increasing the amount of Tween 20 and gelatin to increase the stability of citral, it had a better effect [35]. When tea polyphenol was added at 1%, the citral degradation rate was the lowest, and the chemical stability of the citral was the highest. Therefore, tea polyphenol should be added at 1.0% to reduce citral degradation.

#### 3.2.3. Effect of Citral Concentration on its Degradation Performance

Based on the aforementioned results, the degradation performance of citral at different soluble concentrations was investigated with the addition of tea polyphenol at 1%, and the results are shown in Figure 4.

The citral degradation rate gradually decreased with an increase in the citral solution concentration (Figure 4). When the citral concentration was 125 mg/mL, the citral degradation rate was 5.64%, and when the concentration was 150 mg/mL, the degradation rate was 4.32%, which was 76.60% of the citral concentration of 125 mg/mL. When the citral concentration was 175 mg/mL, the citral degradation rate was 4.11%, which was 76.60% of the citral concentration of 125 mg/mL and 95.14% of the citral concentration of 150 mg/mL, respectively. When the citral concentration was 150 and 175 mg/mL, the degradation rates at these concentrations did not differ considerably. Therefore, the citral solution with a concentration of 175 mg/mL was considered to show good stability. The citral concentration had little effect on its degradation performance; however, oxidative degradation may be inhibited at higher citral concentrations.

### 3.3. Effect of Antioxidants on the DPPH Radical-Scavenging Rate of Citral

The effects of the addition of tea polyphenols, ascorbic acid, and theaflavin on the DPPH radical-scavenging rate of citral are shown in Figure 5.

The effects of different natural antioxidants on the scavenging rate of DPPH free radicals in the citral solutions differed significantly (Figure 5). As shown in Figure 5a, 0.6 and 1.0% of tea polyphenols reduced the DPPH radical-scavenging rate by 9.05 and 15.35%, respectively, indicating a smaller reduction. As shown in Figure 5b, 0.6 and 1.0% of ascorbic acid reduced the scavenging rate by 39.81 and 57.32%, respectively, indicating a more significant reduction. In Figure 5c, 0.6 and 1.0% theaflavin reduced the radical scavenging rate by 29.58 and 13.88%, respectively, indicating a nonsignificant reduction. All the three natural antioxidants had a good scavenging effect on DPPH radicals in the citral solutions (scavenging rate: >50%). However, with an increase in storage time, the three natural antioxidants decreased the DPPH radical-scavenging rate on the 28th day of storage, with the scavenging rate decreasing most significantly for ascorbic acid. This result was similar to that of Shi et al. [29], who used ε-PL combined with citral to study its antioxidant activity. This indicated that the storage time has less of an effect on the antioxidant properties of ascorbic acid for enhancing citral solution, which was directly related to the susceptibility of ascorbic acid to deterioration and loss of bioactivity under the influence of light and temperature. A comparison of Figure 5a and Figure 5c revealed that the DPPH scavenging rate with theaflavin was similar to that with 0.6% tea polyphenol only at 1% addition, which indicated that the TP-C solution had great scavenging power for DPPH radicals in the solution after storage, and tea polyphenol could reduce citral oxidation to a great extent. Therefore, tea polyphenols were better natural antioxidants for preventing citral degradation.

### 3.4. FT-IR Analysis of Antioxidant-Modified Citral

The results of the FT-IR analysis of bamboo, TP-C bamboo, citral and TP-C solutions on days 1 and 28 of storage are shown in Figure 6 and Figure 7.

As shown in Figure 6, 2928 cm^−1^ was mainly produced by the antisymmetric stretching vibration of saturated -CH3, 2844 cm^−1^ by the symmetric stretching vibration of saturated -CH2, and the absorption enhancement of bamboo chips after treatment was at these two locations [36]. The absorption enhancement at 1736 cm^−1^ was mainly produced by the stretching vibration of C=O. The absorption at 1603 cm^−1^ and 1514 cm^−1^ was produced by the skeletal stretching vibration of the benzene ring [37]. The enhanced absorption at 2928 cm^−1^ and 2844 cm^−1^ was attributed to the molecular groups of the antioxidant modified citral. The enhanced absorption at 1736 cm^−1^ was attributed to the entry of the antioxidant-modified citral into the bamboo chips. The absorption at 1603 cm^−1^ and 1514 cm^−1^ was attributed to the oxidative degradation of the lignin and part of the citral in the bamboo itself. The above indicated that the entry of the antioxidant-modified citral into the bamboo chips did not change the chemical structure of the bamboo material, and the presence of citral and its oxidative degradation molecular groups were found in the bamboo chips.

As shown in Figure 7, 2970 cm^−1^ is the absorption peak of saturated -CH3 stretching vibration and 2810 cm^−1^ is the absorption peak of aldehyde (CHO) stretching vibration. Accordingly, 1670 cm^−1^ is the absorption peak of unsaturated C=O stretching vibration, which is also the characteristic absorption peak of citral [15]. The intensity of this absorption peak appeared weakened, indicating the oxidation of the carbonyl structure on citral, while 1440 cm^−1^ is the absorption peak of mononuclear aromatic C=C skeleton stretching vibration. The absorption peak of the C=C skeleton stretching vibration of mononuclear aromatic hydrocarbons may be formed due to the absorption peak of aromatic hydrocarbons formed after the oxidative degradation of the citral. Compared with the intensity of the absorption peak of the citral solution, the intensity of the absorption peak of the TP-C solution decreased to a certain extent on day 1 of storage and the intensity decreased substantially on day 28 of storage. Compared with the intensity of the absorption peak of the citral solution, the intensity of the absorption peak of the TP-C solution did not change on days 1 and 28 of storage. This was mainly because, in the TP-C solution, the oxidation of tea polyphenol was higher than that of citral, and so it was protected and maintained its original chemical stability. Therefore, the citral solution without any natural antioxidants exhibited more citral degradation, resulting in the weakening of the intensity of its absorption peak [20]. Thus, tea polyphenols could effectively inhibit the oxidative degradation of the citral and maintain the stability of the citral to some extent.

### 3.5. Efficacy of Antioxidant-Modified Citral against Bamboo Mildew

The aforementioned findings revealed tea polyphenol to be a promising natural antioxidant for preventing the oxidative degradation of citral, and the suitable amount of tea polyphenol to achieve this result was 1%. TP-C solutions were prepared at 150, 175, and 200 mg/mL concentrations and pH 6.0. The effects of different concentrations of TP-C solutions on the efficacy of the treatment of bamboo against mildews produced by *PC*, *TV*, *AN*, and HUN were investigated, and the results are presented in Table 2 and Table 3 and Figure 8.

As shown in Table 2 and Table 3, the efficacy of TP-C against bamboo mildew was much higher than that of citral against bamboo mildew. As shown in Table 2, at 150 mg/mL, the efficacy of the TP-C treatment against *AN*, *TV*, and HUN was 90, 100, and 100%, respectively. As shown in Table 3, the efficacy of the citral treatment against the four mildews was 33.25, 66.75, and 16.75% [11], respectively, which was only 36.94, 66.75, and 16.75% of the former, but the efficacy against *PC* was 7.5% lower, as shown in Table 2. When the concentration was 175 mg/mL, the effectiveness of the TP-C antimildew treatment against all four mildews reached 100%, while the effectiveness of the citral antimildew treatment against all four mildews reached 100% only when the citral concentration reached 200 mg/mL. Therefore, with the same control efficacy, tea polyphenols could significantly improve the antimildew treatment performance of citral and reduce the amount of citral used, thus saving costs.

As shown in Figure 8, at 150 mg/mL, only a few mycelia were observed on the surface of the TP-C-treated bamboo [Figure 8(a_1_)], whereas more mycelia and fungal spots were observed on the surface of the citral-treated bamboo [Figure 8(b_1_)]. At 175 mg/mL, no mycelia and fungal spots were observed on the surface of the TP-C-treated bamboo [Figure 8(a_2_)], whereas some mycelia and mycorrhizal spots were observed on the surface of the citral-treated bamboo [Figure 8(b_2_)]. When the concentration was 200 mg/mL, no mycelia and mycorrhizal spots were observed on the surface of the TP-C and citral-treated bamboo [Figure 8(a_3_) and Figure 8(b_3_), respectively]. These findings confirmed that, with the same control efficacy, tea polyphenols could significantly improve the antimildew performance of citral and reduce the amount of citral required for treatment, thereby saving costs.

## 4. Conclusions

In this work, citral modification with antioxidants reduced the amount of citral required in the treatment of bamboo mildew, thereby reducing the treatment costs for bamboo mildew. With the addition of natural antioxidants such as tea polyphenols, theaflavins, and ascorbic acid to citral, the citral degradation rate decreased for all types of antioxidants, and the scavenging rate of DPPH free radicals in the citral solution was high. Natural antioxidants could improve the antioxidant performance of citral to some extent. Among these antioxidants, tea polyphenols had the best performance and could effectively prevent citral oxidative degradation. The citral degradation rate decreased significantly with an increase in the tea polyphenol content. The citral concentration had little effect on its degradation performance, and tea polyphenol should be added at 1.0% to reduce citral degradation. Tea polyphenols could effectively inhibit the oxidative degradation of citral and maintain its stability to a certain extent. The efficacy of TP-C against all four species of mildews in the bamboo antimildew treatment reached 100% at 175 mg/mL; however, the effectiveness of citral against the mildew species reached 100% when its concentration was 200 mg/mL. Therefore, the dosage of citral could be reduced by 25 mg/mL for the same efficacy, saving on treatment costs and improving citral stability at the same time. Tea-polyphenol-modified citral can improve the antioxidant properties of citral, and it has a longer anti-mildew time after being impregnated on bamboo, which is beneficial to the preservation of bamboo and its products.

## Figures and Tables

**Figure 1 polymers-14-04652-f001:**
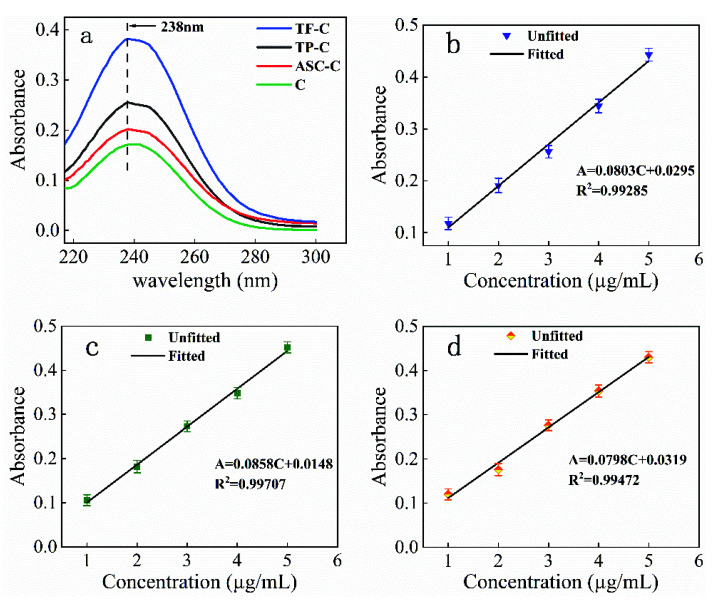
UV absorption spectrum of antioxidant-modified citral. (**a**) UV spectra of citral, tea-polyphenol–citral, ascorbic-acid–citral, theaflavin–citral, (**b**) standard curves of tea-polyphenol–citral, (**c**) standard curves of ascorbic-acid–citral, (**d**) standard curves of theaflavin–citral.

**Figure 2 polymers-14-04652-f002:**
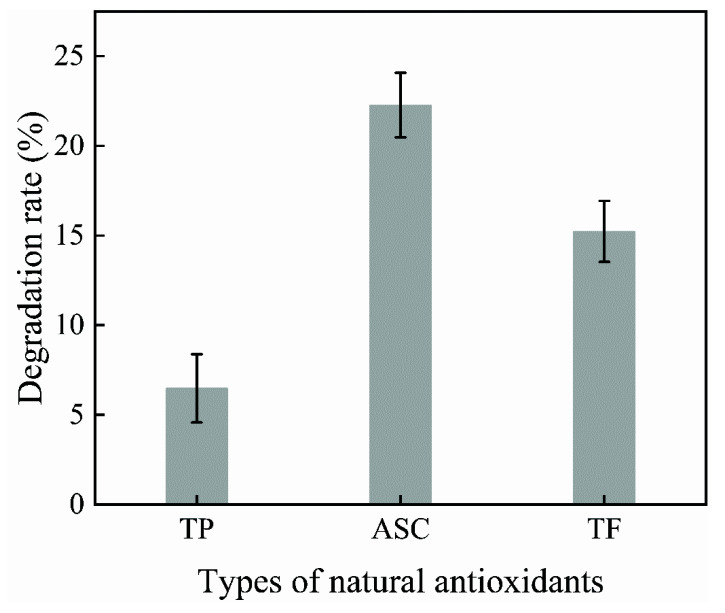
Effect of 1.0% natural antioxidant species on the citral degradation rate.

**Figure 3 polymers-14-04652-f003:**
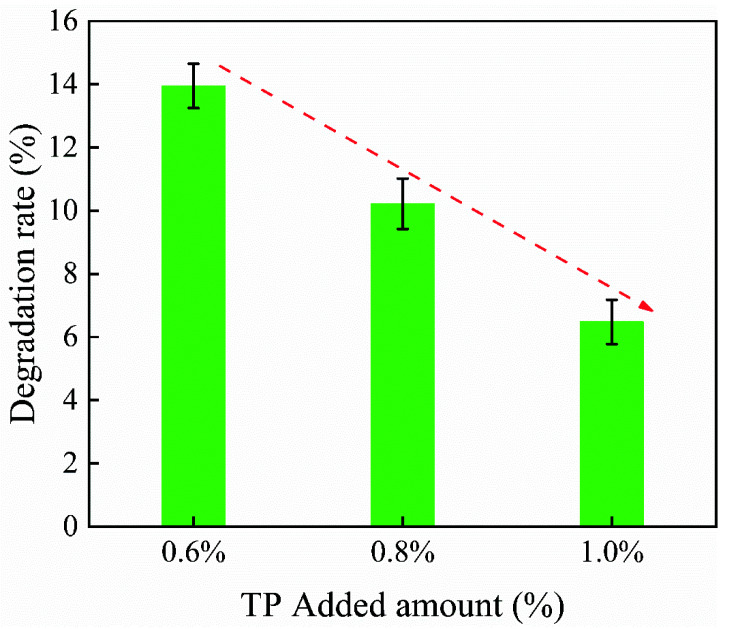
Effect of antioxidant addition on the citral degradation rate.

**Figure 4 polymers-14-04652-f004:**
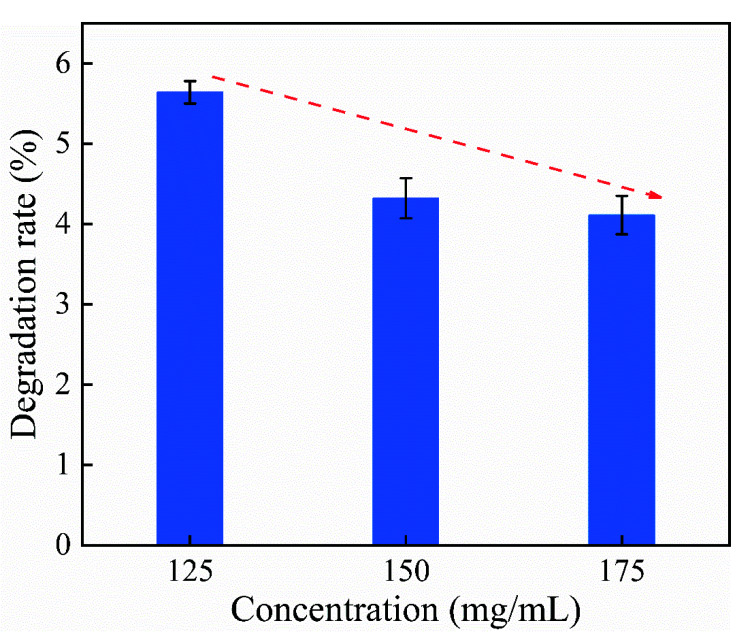
Effect of citral concentration on its degradation rate.

**Figure 5 polymers-14-04652-f005:**
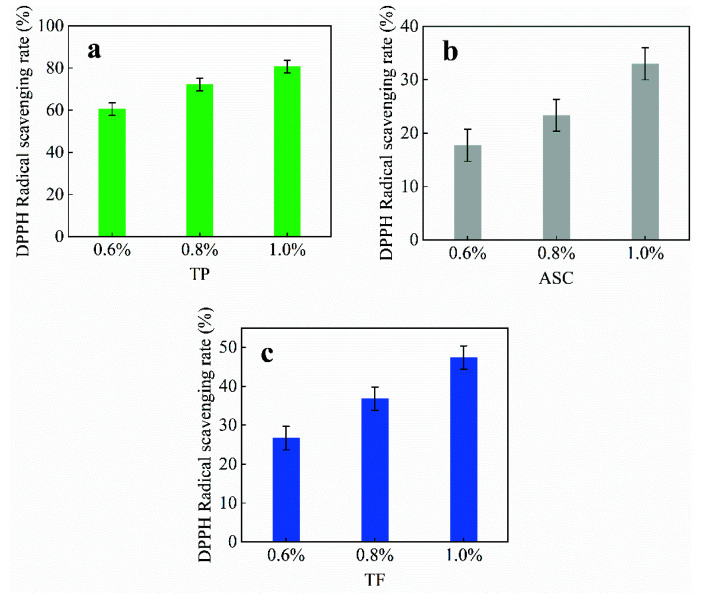
Effect of natural antioxidant species on the DPPH radical-scavenging rate of citral. (**a**): the DPPH radical scavenging rate of tea polyphenols; (**b**): the DPPH radical scavenging rate of ascorbic acid; (**c**): the DPPH radical scavenging rate of theaflavins.

**Figure 6 polymers-14-04652-f006:**
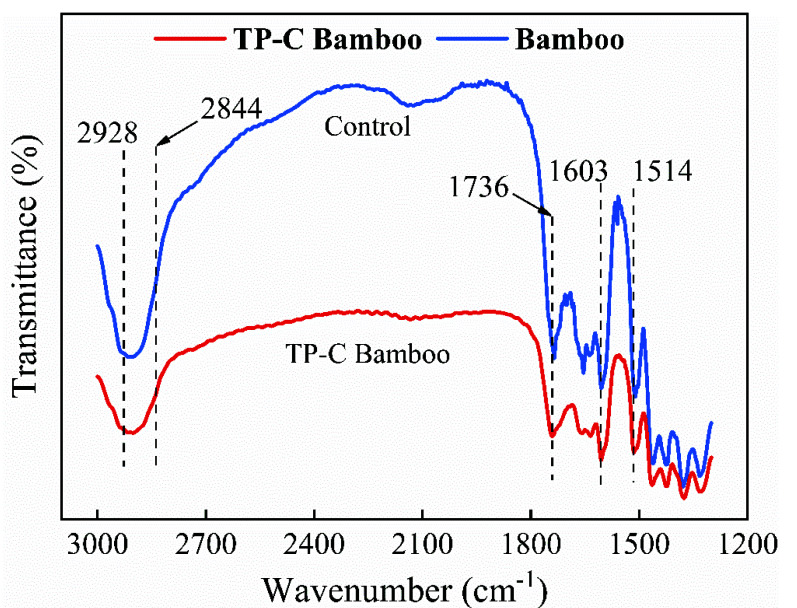
FT–IR infrared spectra of bamboo and antioxidant-modified citral bamboo.

**Figure 7 polymers-14-04652-f007:**
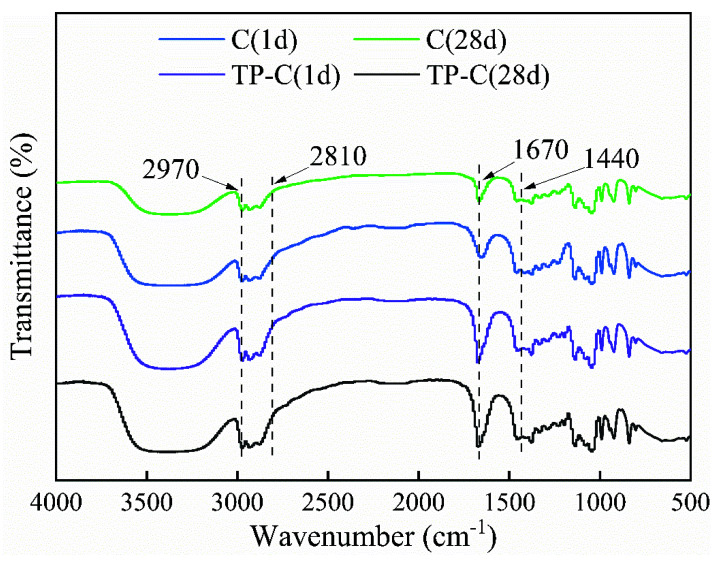
FT–IR infrared spectra of citral and tea-polyphenol–citral.

**Figure 8 polymers-14-04652-f008:**
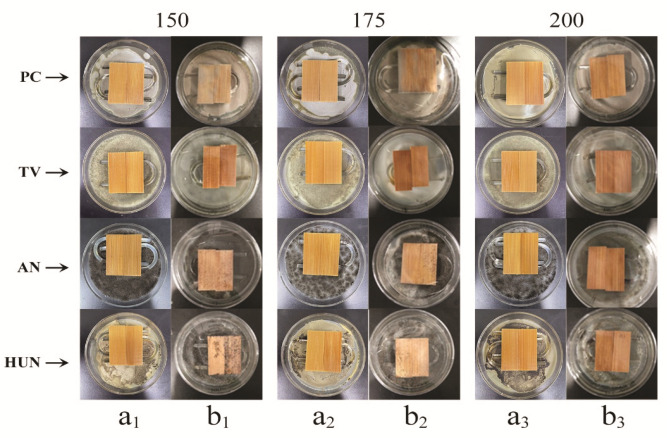
Comparison of the antimildew effect of different concentrations of tea-polyphenol-modified citral and citral antimildew treatment on bamboo. (**a_1_**): 150 mg/mL tea-polyphenol citral treated bamboo, (**a_2_**): 175 mg/mL tea-polyphenol citral treated bamboo, (**a_3_**): 200 mg/mL tea-polyphenol citral treated bamboo; (**b_1_**): 150 mg/mL citral treated bamboo, (**b_2_**): 175 mg/mL citral treated bamboo, (**b_3_**): 200 mg/mL citral treated bamboo.

**Table 1 polymers-14-04652-t001:** The grading standard of the surface infection value of mildews for samples.

Infection Value	Infected Area of Specimen
0	No mycelium or mildew on the specimen surface
1	Infected area of specimen surface <1/4
2	Infection area of specimen surface 1/4 to 1/2
3	Infection area of specimen surface 1/2 to 3/4
4	Infection area of specimen surface >3/4

**Table 2 polymers-14-04652-t002:** Efficacy of different concentrations of tea-polyphenols-citral on the mildew treatment of bamboo.

Concentration (mg/mL)	Antimildew Efficiency (%)			
	*AN*	*PC*	*TV*	HUN
150	90	92.5	100	100
175	100	100	100	100
200	100	100	100	100

**Table 3 polymers-14-04652-t003:** Efficacy of different concentrations of citral on the mildew treatment of bamboo [11].

Concentration (mg/mL)	Antimildew Efficiency (%)			
	*AN*	*PC*	*TV*	HUN
150	33.25	100	66.75	16.75
175	41.75	100	83.25	66.75
200	100	100	100	100

## Data Availability

Not applicable.

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
