# Peer review of "Antioxidant-Mediated Modification of Citral and Its Control Effect on Mildewy Bamboo"

_polymers, 2022, doi:10.3390/polym14214652_

Round 1
Reviewer 1 Report
The topic of the manuscript "Antioxidant-Mediated Modification of Citral and Its Control Effect on Mildewy Bamboo" is arresting and will be of interest to the readers of the journal.
In my opinion, the Introduction should be expanded. For example, in "The damage caused is so profound that bamboo loses its use value and is discarded, resulting in a significant waste of resources and economic loss [6-10]" (lines 29-31), five sources are cited. In "To improve citral stability, domes-43 tic and foreign scholars have used microcapsules to encapsulate citral and to encapsulate citral with antioxidants [11, 12, 23-30]" (lines 43-45), ten references are cited in just one sentence the source etc. It would be good if the respected authors would give in more detail the main results from the cited sources.
The materials and methods are well described, but it would be good to increase the justification of the parameters used. That is, to give more references from similar studies, based on which to justify the study's parameters.
The analysis of the results is detailed, but in practice, there is no comparison with similar studies. I ask the authors to improve the quality of figures 1 to 5. That is, please provide these graphs with a higher resolution. There is a mistake in figure 4: "degradation….", and it should be with capital D.
I ask the authors to consider whether the conclusion should be given as separate sub-points. In my opinion, the conclusion should include the study's main contribution and its main novelty.
Please complete the sentence: "Supplementary Materials: The following are available online at" (line 341).
Overall, in my opinion, the research is interesting and qualitatively conducted, but for the manuscript to be suitable for publication in Polymers, its quality should be improved.
Author Response
Point 1: In my opinion, the Introduction should be expanded. For example, in "The damage caused is so profound that bamboo loses its use value and is discarded, resulting in a significant waste of resources and economic loss [6-10]" (lines 29-31), five sources are cited. In "To improve citral stability, domes-43 tic and foreign scholars have used microcapsules to encapsulate citral and to encapsulate citral with antioxidants [11, 12, 23-30]" (lines 43-45), ten references are cited in just one sentence the source etc. It would be good if the respected authors would give in more detail the main results from the cited sources.
Response 1: Thank you for your valuable and helpful comment. According to the your comment, We have deleted unnecessary references in the background part of the introduction in the paper, and supplemented the content on the original basis. The content is revised as follows:
The damage caused is so profound that bamboo loses its use value and is discarded, resulting in a significant waste of resources and economic loss[6-7].
- Nirmala, C.; Bisht, M.S.; Bajwa, H.K.; Santosh, O. Bamboo: A rich source of natural antioxidants and its applications in the food and pharmaceutical industry. Trends in Food Science & Technology 2018, 77, 91-99.
- Liu, W.; Hui, C.; Wang, F.; Wang, M.; Liu, G. Review of the Resources and Utilization of Bamboo in China. Bamboo-Current and Future Prospects 2018, 133-142.
To improve citral stability, Ju[8], Chang[18], Qiu[19], Wang[20], Lu[21], et al, domestic and Arnon-Rips[22], Mendes[23], Sharma[24] et al, foreign scholars have used microcapsules to encapsulate citral and to encapsulate citral with antioxidants.
- Ju, J.; Xie, Y.; Yu, H.; Guo, Y.; Cheng, Y.; Qian, H.; Yao, W. Analysis of the synergistic antifungal mechanism of eugenol and citral. LWT 2020, 123, 109128.
18 Chang, X.; Chen, C.; Gong, D.; Dong, Q. Research progress of natural antioxidants to inhibit oxidation of fats and oils. China Fats and Oils 2020, 45, 46-50.
- Qiu, B.; Zhou, Y.; Yin, X.; Chen, J.; Yin, Y.; Zhu, L. Preparation of citral microcapsules by spray drying method and study of microcapsule stability. Food Industry Science and Technology 2017, 38, 190-195.
- Wang, G.; Zhao, H.; Chen, X.; Li, W.; Song, S. Inhibition of citral degradation by plant-based natural antioxidants. Flavors and Cosmetics 2019, 29-32.
- Lu, W.-C.; Huang, D.-W.; Wang, C.-C.; Yeh, C.-H.; Tsai, J.-C.; Huang, Y.-T.; Li, P.-H. Preparation, characterization, and antimicrobial activity of nanoemulsions incorporating citral essential oil. Journal of food and drug analysis 2018, 26, 82-89.
- Arnon-Rips, H.; Sabag, A.; Tepper-Bamnolker, P.; Chalupovich, D.; Levi-Kalisman, Y.; Eshel, D.; Porat, R.; Poverenov, E. Effective suppression of potato tuber sprouting using polysaccharide-based emulsified films for prolonged release of citral. Food Hydrocolloid 2020, 103, 105644.
- Mendes, J.; Norcino, L.; Martins, H.; Manrich, A.; Otoni, C.; Carvalho, E.; Piccoli, R.; Oliveira, J.; Pinheiro, A.; Mattoso, L. Correlating emulsion characteristics with the properties of active starch films loaded with lemongrass essential oil. Food Hydrocolloid 2020, 100, 105428.
- Sharma, K.; Guleria, S.; Razdan, V.K.; Babu, V. Synergistic antioxidant and antimicrobial activities of essential oils of some selected medicinal plants in combination and with synthetic compounds. Ind Crop Prod 2020, 154, 112569.
Point 2: The materials and methods are well described, but it would be good to increase the justification of the parameters used. That is, to give more references from similar studies, based on which to justify the study's parameters.
Response 2: Thank you for your valuable and helpful comment. According to the your comment, We have added relevant references to the materials and methods involved in the paper, and the contents are modified as follows:
A propylene glycol solution (pH: 6.0) was prepared at room temperature[29].
- Shi, C.; Zhao, X.; Liu, Z.; Meng, R.; Chen, X.; Guo, N. Antimicrobial, antioxidant, and antitumor activity of epsilon-poly-L-lysine and citral, alone or in combination. Food & nutrition research 2016, 60, 31891.
Then, 1.0 g of citral and 0.01 g of natural antioxidants (theaflavin, tea polyphenols, and ascorbic acid) were weighed[30]
- Yang, X.; Tian, H.; Ho, C.-T.; Huang, Q. Inhibition of citral degradation by oil-in-water nanoemulsions combined with antioxidants. Journal of agricultural and food chemistry 2011, 59, 6113-6119.
1,1-Diphenyl-2-trinitrophenylhydrazyl radical 2,2-diphenyl-1-(2,4,6-trinitrophenyl) hydrazyl, DPPH] solution was prepared[31].
- Kim, S.H.; Song, H.Y.; Choi, S.J. Influence of structural properties of emulsifiers on citral degradation in model emulsions. Food science and biotechnology 2019, 28, 701-710.
Point 3: The analysis of the results is detailed, but in practice, there is no comparison with similar studies. I ask the authors to improve the quality of figures 1 to 5. That is, please provide these graphs with a higher resolution. There is a mistake in figure 4: "degradation….", and it should be with capital D.
Response 3: Thank you for your valuable and helpful comment. According to the your comment, We have added other researchers' comparative analysis of citral anti-mildew and antibacterial in the section of results analysis, and added corresponding references. Thank you for your reminder. We have modified Figure 1-5 of the article results, changed the error of "degradation" to "Degradation", and output a higher resolution picture. The result analysis part is modified as follows:
Figure 1. UV absorption spectrum of antioxidant-modified citral. (a) UV spectra of citral, tea polyphenol–citral, ascorbic acid–citral, theaflavin–citral, (b) Standard curves of tea polyphenol–citral, (c) standard curves of ascorbic acid–citral, (d) Standard curves of theaflavin–citral.
Figure 2. Effect of 1.0% natural antioxidant species on the citral degradation rate.
Figure 3. Effect of antioxidant addition on the citral degradation rate.
Figure 4. Effect of citral concentration on its degradation rate.
Figure 5. Effect of natural antioxidant species on the DPPH radical scavenging rate of citral.
Point 4: I ask the authors to consider whether the conclusion should be given as separate sub-points. In my opinion, the conclusion should include the study's main contribution and its main novelty.
Response 4: Thank you for your valuable and helpful comment. According to the your comment, We combined the point-by-point conclusions in the paper into one paragraph, and reorganized the conclusions of the paper. The changes in the conclusion part of the paper are as follows:
In this work, Citral modification with antioxidants reduced the amount of citral required in the treatment, thereby reducing the treatment cost for bamboo mildew. With the addition of natural antioxidants such as tea polyphenols, theaflavins, and ascorbic acid to citral, the citral degradation rate decreased for all types of antioxidants, and the scavenging rate of DPPH free radicals in the citral solution was high. Natural antioxi-dants can improve the antioxidant performance of citral to some extent. Among these antioxidants, tea polyphenols have the best performance and can effectively prevent citral oxidative degradation. The citral degradation rate decreased significantly with an increase in the tea polyphenol content. The citral concentration had little effect on its degradation performance, and tea polyphenol should be added at 1.0% to reduce citral degradation. Tea polyphenols can effectively inhibit the oxidative degradation of citral and maintain its stability to a certain extent. The efficacy of TP-C against all four species of mildews in the bamboo antimildew treatment reached 100% at 175 mg/mL; however, the effectiveness of citral against the mildew species reached 100% when its concen-tration was 200 mg/mL. Therefore, the dosage of citral can be reduced by 25 mg/mL for the same efficacy, saving the treatment cost and improving citral stability at the same time. The tea polyphenol-modified citral can improve the antioxidant properties of citral, and it has a longer anti-mildew time after being impregnated on bamboo. The tea pol-yphenol-modified citral is beneficial to the preservation of bamboo and its products.
Point 5: Please complete the sentence: "Supplementary Materials: The following are available online at" (line 341).
Response 5: Thank you very much for your careful review. We have completed the sentences as required, and uploaded the relevant materials of the article to the materials submission website of Polymers. The links for supplementary materials are as follows:
Point 6: Overall, in my opinion, the research is interesting and qualitatively conducted, but for the manuscript to be suitable for publication in Polymers, its quality should be improved.
Response 6: Thank you for your valuable and helpful comment. According to your comment, We have carefully revised and supplemented the FTIR-related experimental data (see line 275-289) of the combination of citral and bamboo after the antioxidant-modified citral was pressure-impregnated with bamboo to suit the purpose and publication requirements of the Polymers journal.

Reviewer 2 Report
The paper is interesting, some comments for improvement are as below:
Line 36 – Please italicize Cyperus rotundus
Line 57 – “Therefore, this study discusses the antimildew properties of citral”, the focus should be on the effects of natural antioxidants on the efficacy of citral. Please rewrite the objective of the study (Line 57-59) to better reflect the content of the study.
Line 169 – “wavelength range (a)”, please write (Figure 1a) in the bracket instead of just putting (a).
Figure 1b, the x-axis is bolded, please unbold.
Line 179 – “R2” – superscript please
Figure 2 – why the degradation of pure citral was not presented? Please mention in the figure title how many % antioxidant was added and at what concetration
Line 222 – “Therefore, tea polyphenol should be added at 1.0% to reduce citral degradation.” How about addition of more than 1.0%?
Figure 3 – what concentration was used here?
Line 233 – “When the citral concentration was 175 mg/mL, the citral degradation rate was 4.11%, which was 76.60% and 95.14% of the citral concentrations of 125 and 150 mg/mL, respectively” I am not quite sure about this statement, please clarify what do you mean?
Line 235 – “When the citral concentration was 150 and 175 mg/mL, the degradation rates at these concentrations did not differ considerably” since they did not differ considerably, what not the authors chose a lower concentration of 150 mg/mL? please provide justification
Figure 5 – DPPH scavenging rate or clearance rate? Please use it consistently.
Line 246 – can you report the DPPH scavenging rate of each type of antioxidants first before comparing their reduction?
Can you combine Table 2 & 3 for better comparison?
Line 323 – regrading cost saving, out of curiosity, is the price of antioxidant far lower than citral?
A more in-depth discussion is required, more references and previous studies are beneficial.
Author Response
Point 1: Line 36 – Please italicize Cyperus rotundus.
Response 1: Thank you very much for your careful review. We have changed "Cyperus rotundus" to "Litsea cubeba" in line 36 of the paper and used italics to modify the content as follows:
Among these agents, citral, a green, non-toxic natural antimicrobial agent mainly derived from Litsea cubeba essential oil, has broad-spectrum antimicrobial activity and strong lemon fragrance.
Point 2: Line 57 – “Therefore, this study discusses the antimildew properties of citral”, the focus should be on the effects of natural antioxidants on the efficacy of citral. Please rewrite the objective of the study (Line 57-59) to better reflect the content of the study.
Response 2: Thank you very much for your careful review. We make the following changes to lines 57-59 in the introduction:
Therefore, this study discusses the efficacy of natural antioxidants on the antimildew properties of citral. The research results lay the foundation and provide a theoretical reference for the promotion and application of natural antioxidant modified citral as an antimildew agent for bamboo.
Point 3: Line 169 – “wavelength range (a)”, please write (Figure 1a) in the bracket instead of just putting (a).
Response 3: Thank you very much for your careful review. We revised the incorrect statement of "wavelength range (a)" in line 169 of the paper to "wavelength range (Figure 1a)", and correspondingly revised "wavelength (b, c, d)" of 171 to "wavelength (Figure 1)"
Point 4: Figure 1b, the x-axis is bolded, please unbold.
Response 4: Thank you very much for your careful review. We unbold the X-axis in Figure 1b, and the modified results are as follows:
Figure 1. UV absorption spectrum of antioxidant-modified citral. (b) Standard curves of tea polyphenol–citral
Point 5: Line 179 – “R2” – superscript please
Response 5: Thank you very much for your careful review. We changed "R2" in line 179 of the paper to "R2", and the result of the modification is as follows:
A = 0.0803C + 0.0295, R2 = 0.99285. Figure 1(c) shows the standard curve regression equation of ASC-C solution, A = 0.0858C + 0.0148, R2 = 0.99707. Figure 1(d) shows the standard curve regression equation of TF-C solution, A = 0.0798C + 0.0319, R2 = 0.99472. The fitted correlation coefficient of the aforementioned regression equation, R2, was very close to 1, indicating that the concentration of antioxidant-modified citral had a good linear relationship with absorbance.
Point 6: Figure 2 – why the degradation of pure citral was not presented? Please mention in the figure title how many % antioxidant was added and at what concetration.
Point 6: Figure 2 – why the degradation of pure citral was not presented? Please mention in the figure title how many % antioxidant was added and at what concetration.
Response 6: Thank you for your valuable and helpful comment. We consider that pure citral itself has a greater degree of degradation under the same conditions compared to citral without antioxidants, and Figure 2 mainly compares the slowing degree of three natural antioxidants on citral degradation. In addition, we supplemented the addition amount of each natural antioxidant in the title of Figure 2. In the experiment, the addition amount of natural antioxidant was 10% of the mass of citral, and a certain amount of citral of the corresponding concentration was taken to dissolve. In the case of the same solution concentration, the total volume of the solution is constant, and the amount of natural antioxidants added is fixed. It is not meaningful to calculate the concentration of natural antioxidants in the solution, and the final antioxidant modified citral solution exists in the solution. For various components, there is a relatively large error in calculating the concentration of natural antioxidants, so the experiment is calculated based on the addition percentage of natural antioxidants. The results of the modification of the title in Figure 2 are as follows:
Figure 2. Effect of 1.0% natural antioxidant species on the citral degradation rate.
Point 7: Line 222 – “Therefore, tea polyphenol should be added at 1.0% to reduce citral degradation.” How about addition of more than 1.0%?
Response 7: Thank you for your valuable and helpful comment. In the experiment of natural antioxidant modified citral, we used the method of equal difference increase to explore the effect of the addition of natural antioxidant tea polyphenols on the degradation rate of citral. The effect of the addition of tea polyphenols from 06% to 1.0% on the degradation rate of citral has been on a downward trend. When polyphenols were added in an amount of 1.0%, they already had a good antioxidant protection effect on citral. Continuing to increase the amount of tea polyphenols had little significance in reducing the oxidative degradation of citral. Considering the high price of high-purity tea polyphenols, increasing the amount of tea polyphenols will increase the modification cost of citral. The addition of tea polyphenols above 1.0% will be further explored in subsequent related experiments.
Point 8: Figure 3 – what concentration was used here?
Response 8: Thank you very much for your careful review. In exploring the effect of tea polyphenol addition on the degradation rate of citral, the concentration of citral solution used for different tea polyphenol additions was 150 mg/mL, and the tea polyphenols were added at 10% of the mass of citral, expressed as a percentage.
Point 9: Line 233 – “When the citral concentration was 175 mg/mL, the citral degradation rate was 4.11%, which was 76.60% and 95.14% of the citral concentrations of 125 and 150 mg/mL, respectively” I am not quite sure about this statement, please clarify what do you mean?
Response 9: Thank you for your valuable and helpful comment. We have made corresponding changes to the vague sentence in the comparison of the citral degradation rate results in line 233 of the text, and the changes are as follows:
When the citral concentration was 175 mg/mL, the citral degradation rate was 4.11%, which was 76.60% of the citral concentrations of 125 mg/mL and 95.14% of the citral concentrations of 150 mg/mL, respectively.
Point 10: Line 235 – “When the citral concentration was 150 and 175 mg/mL, the degradation rates at these concentrations did not differ considerably” since they did not differ considerably, what not the authors chose a lower concentration of 150 mg/mL? please provide justification.
Response 10: Thank you for your valuable and helpful comment. From the results of the degradation rate of antioxidant modified citral alone, the antioxidant citral solution with a concentration of 150 mg/mL has a lower degradation rate, and the degradation rate is comparable to that of the antioxidant citral solution with a concentration of 175 mg/mL. In comparison, there is not much difference between the two. After 28 days of anti-fungal experiments on bamboo, it was concluded that the anti-oxidative citral with a concentration of 150 mg/mL has a good anti-fungal effect, but its anti-fungal effect on bamboo has not yet reached 100%, and the surface of bamboo is still growing. Mycelium and plaque, while the antioxidant citral at a concentration of 175 mg/mL was 100% effective against mildew on bamboo. Therefore, the antioxidant modified citral with a concentration of 175 mg/mL was finally selected for bamboo mildew control under the condition that the difference between the concentrations of 150 mg/mL and 175 mg/mL on the degradation rate of citral was small.
Point 11: Figure 5 – DPPH scavenging rate or clearance rate? Please use it consistently.
Response 11: Thank you very much for your careful review. We have kept the ordinate title "DPPH clearance rate" in Figure 5 consistent with the title in Figure 5, and modified it to "DPPH Radical scavenging rate". The modification results are as follows:
Figure 5. Effect of natural antioxidant species on the DPPH radical scavenging rate of citral.
Point 12: Line 246 – can you report the DPPH scavenging rate of each type of antioxidants first before comparing their reduction?
Response 12: Thank you for your valuable and helpful comment. We have modified the expression of DPPH free clearance as follows:
As shown in Figure 5(a), 0.6% and 1.0% of tea polyphenols the DPPH radical scavenging rate were 60.5% and 80.77%, respectively, and their reductions were 9.05% and 15.35%, respectively, indicating a smaller reduction. As shown in Figure 5(b), 0.6% and 1.0% of ascorbic acid the DPPH radical scavenging rate were 17.72% and 32.96%, respectively, and their reductions were 39.81% and 57.32%, respectively, indicating a more significant reduction. In Figure 5(c), 0.6% and 1.0% theaflavin the DPPH radical scavenging rate were 26.67% and 47.38%, respectively, and their reductions were 29.58% and 13.88%, respectively, indicating a nonsignificant reduction. All the three natural antioxidants had a good scavenging effect on DPPH radicals in citral solution (scavenging rate: >50%).
Point 13: Can you combine Table 2 & 3 for better comparison?
Response 13: Thank you for your valuable and helpful comment. We have combined Table 2 and Table 3 for comparative analysis, and the revised results are as follows:
Table 2. Efficacy of different concentrations of tea polyphenols-citral against the mildew treatment of bamboo.
|
Concentration (mg/mL) |
Antimildew efficiency (%) |
|||
|
AN |
PC |
TV |
HUN |
|
|
150 |
90 |
92.5 |
100 |
100 |
|
175 |
100 |
100 |
100 |
100 |
|
200 |
100 |
100 |
100 |
100 |
Table 3. Efficacy of different concentrations of citral against the mildew treatment of bamboo
|
Concentration (mg/mL) |
Antimildew efficiency (%) |
|||
|
AN |
PC |
TV |
HUN |
|
|
150 |
33.25 |
100 |
66.75 |
16.75 |
|
175 |
41.75 |
100 |
83.25 |
66.75 |
|
200 |
100 |
100 |
100 |
100 |
As shown in Tables 2 and 3, the efficacy of TP-C against bamboo mildew was much higher than that of citral against bamboo mildew. As shown in Table 2, at 150 mg/mL, the efficacy of TP-C treatment against AN, TV, and HUN was 90%, 100%, and 100%, respectively. As shown in Table 3, the efficacy of citral treatment against the four mildews was 33.25%, 66.75%, and 16.75%[11], respectively, which was only 36.94%, 66.75% and 16.75% of the former. 66.75%, 16.7%, but the efficacy against PC was 7.5% lower in Table 2.
Point 14: Line 323 – regrading cost saving, out of curiosity, is the price of antioxidant far lower than citral?
Response 14: Thank you for your valuable and helpful comment. Both tea polyphenols and citral are natural extracts, and the cost is not low. The price of natural antioxidants of tea polyphenols is not much lower than that of citral. The market prices of the two are 99% of tea polyphenols 25. CNY/g, 97% citral is 1.6 CNY/mL. In terms of price, there is a big price difference between the two, but as a natural antioxidant, tea polyphenols only need 10% of citral. Under the same concentration of citral solution, the same anti-fungal effect of bamboo can be achieved. The amount of citral solution without antioxidants needs more, and the amount of oxidation during the anti-fungal process is more, while tea polyphenol antioxidants only A smaller amount is required to slow down the oxidation of citral to a certain extent, while reducing the amount of citral.
Point 15: A more in-depth discussion is required, more references and previous studies are beneficial.
Response 15: Thank you for your valuable and helpful comment. We have added a discussion to the analysis of the results, compared with other similar studies, and added relevant references. The analysis and discussion are revised as follows:
The maximum absorption peaks of TP-C, ASC-C, and TF-C at 238 nm in Figure 1(a) coincide with the absorption peak of pure citral, which indicates that the TP-C, ASC-C, and TF-C solutions were not affected by natural antioxidants or the citral structure was not damaged. The absorption at this wavelength is consistent with the findings of Peng[33] and Ay[34].
- Ay, E.; Gérard, V.; Graff, B.; Morlet-Savary, F.; Mutilangi, W.; Galopin, C.; Lalevée, J. Citral photodegradation in solution: Highlighting of a radical pathway in parallel to cyclization pathway. Journal of agricultural and food chemistry 2019, 67, 3752-3760.
- Peng, R.; Du, C.; Hu, A.; Li, Q.; Zhang, J.; Zhang, W.; Sun, F. Fabrication of core–shell type poly (NIPAm)-encapsulated citral and its application on bamboo as an anti-molding coating. Rsc Adv 2021, 11, 36884-36894.
With the addition of tea polyphenol at 0.6%, the citral degradation rate was 13.95%; with 0.8% tea polyphenol, the degradation rate was 10.22%; and with 1.0% tea polyphenol, the degradation rate was 6.48%. Compared with increasing the amount of Tween 20 and gelatin to increase the stability of citral, it has a better effect[35].
- Tian, H.; Li, D.; Xu, T.; Hu, J.; Rong, Y.; Zhao, B. Citral stabilization and characterization of nanoemulsions stabilized by a mixture of gelatin and Tween 20 in an acidic system. Journal of the Science of Food and Agriculture 2017, 97, 2991-2998.
However, with an increase in storage time, the three natural antioxidants decreased the DPPH radical scavenging rate on the 28th day of storage, with the scavenging rate decreasing most significantly for ascorbic acid. This result is similar to that of Shi et al.[29] using ɛ-PL combined with citral to study its antioxidant activity.
- Shi, C.; Zhao, X.; Liu, Z.; Meng, R.; Chen, X.; Guo, N. Antimicrobial, antioxidant, and antitumor activity of epsilon-poly-L-lysine and citral, alone or in combination. Food and Nutrition Research 2016, 60, 31891.
2928 cm−1 was mainly produced by the antisymmetric stretching vibration of saturated-CH3, 2844 cm−1 by the symmetric stretching vibration of saturated-CH2, and the absorption enhancement of bamboo chips after treatment at these two locations[36].
- Su, N.; Fang, C.; Zhou, H.; Tang, T.; Zhang, S.; Wang, X.; Fei, B. Effect of Rosin Modification on the Visual Characteristics of Round Bamboo Culm. Polymers 2021, 13, 3500.
The absorption enhancement at 1736 cm−1 was mainly produced by the stretching vibration of C=O. The absorption at 1603 cm−1 and 1514 cm−1 was produced by the skeletal stretching vibration of the benzene ring[37].
- Xu, G.; Wang, L.; Liu, J.; Wu, J. FTIR and XPS analysis of the changes in bamboo chemical structure decayed by white-rot and brown-rot fungi. Appl Surf Sci 2013, 280, 799-805.
2970 cm−1 is the absorption peak of saturated-CH3 stretching vibration and 2810 cm−1 is the absorption peak of aldehyde (CHO) stretching vibration. Accordingly, 1670 cm−1 is the absorption peak of unsaturated C=O stretching vibration, which is also the characteristic absorption peak of citral[15].
- Prakash, A.; Baskaran, R.; Vadivel, V. Citral nanoemulsion incorporated edible coating to extend the shelf life of fresh cut pineapples. Lwt - Food Science and Technology 2020, 118, 108851.

Round 2
Reviewer 1 Report
Overall, in my opinion, the manuscript is significantly improved.
The respected authors have complied with the recommendations I have made.
That gives me a reason to recommend that the manuscript be accepted in its present form.
Reviewer 2 Report
The paper has been greatly improved and it is acceptable at the current form